# Thermoelectric Properties of n-Type Molybdenum Disulfide (MoS_2_) Thin Film by Using a Simple Measurement Method

**DOI:** 10.3390/ma12213521

**Published:** 2019-10-26

**Authors:** Shakeel Ashraf, Viviane Forsberg, Claes G. Mattsson, Göran Thungström

**Affiliations:** 1Department of electronics, Mid Sweden University, 851 70 Sundsvall, Swedengoran.thungstrom@miun.se (G.T.); 2Royal Institute of Technology KTH, Wallenberg Wood Science Center, SE-10044 Stockholm, Sweden

**Keywords:** thermoelectricity, thermoelectric generators, Seebeck coefficient, MoS_2_, exfoliated, thin films, green energy harvesting, molybdenum disulfide, molybdenite crystal

## Abstract

In this paper, a micrometre thin film of molybdenum disulfide (MoS_2_) is characterized for thermoelectric properties. The sample was prepared through mechanical exfoliation of a molybdenite crystal. The Seebeck coefficient measurement was performed by generating a temperature gradient across the sample and recording the induced electrical voltage, and for this purpose a simple measurement setup was developed. In the measurement, platinum was utilized as reference material in the electrodes. The Seebeck value of MoS_2_ was estimated to be approximately −600 µV/K at a temperature difference of 40 °C. The negative sign indicates that the polarity of the material is n-type. For measurement of the thermal conductivity, the sample was sandwiched between the heat source and the heat sink, and a steady-state power of 1.42 W was provided while monitoring the temperature difference across the sample. Based on Fourier’s law of conduction, the thermal conductivity of the sample was estimated to be approximately 0.26 Wm^−1^ K^−^. The electrical resistivity was estimated to be 29 Ω cm. The figure of merit of MoS_2_ was estimated to be 1.99 × 10^−4^.

## 1. Introduction

With the development of smart industries, smart cities and the Internet of Things (IoT), the demand for wireless sensor systems has increased rapidly. A major challenge for these systems is continuous operation with a minimized need for maintenance and periodic battery changes. An environmentally friendly solution to the battery problem is to replace the battery with an energy-harvesting solution based on green materials. Heat, or thermal energy, is among the most common sources of waste energy that can be found. As a result, the possibility of harvesting energy thermoelectrically is particularly interesting [1]. The field of thermoelectricity is interesting not only for power generation but also for electronic refrigeration [2]. Evaluation of materials suitable as thermoelectric (TE) generators is greatly determined by the measurement of the figure of merit (ZT) of the material. To have high conversion efficiency, the material should have a large Seebeck coefficient (S) and, simultaneously, low electrical resistivity (*ρ*) and low thermal conductivity (*κ*), as described by Equation (1), where *T* is the temperature:(1)$$ZT=S^{2}T/\rho \kappa$$

The factor ($S^{2}/\rho$) in Equation (1) is known as the TE power factor. For narrow bandgap semiconductors, this factor is optimized with a carrier concentration of around $10^{19}{\;\mathrm{cm}}^{-3}$ [3]. 

After the discovery of graphene [4], the interest in materials that could also be exfoliated due to their layered structure increased dramatically. Since graphene does not have a bandgap, it is not considered as a TE material unless a bandgap is engineered [5]. For this reason, many thermoelectric materials are being explored for power generation applications, such as germanium telluride [6], Silicides [7], half-Heuslers [8], and semiconductor materials such as MoS_2_ have become the topic of numerous publications [9,10,11,12]. MoS_2_ belongs to the class of materials known as transition metal dichalcogenides (TMDs). Due to the quantum confinement, the bandgap of MoS_2_ increases up to 1.9 eV when the thickness of the material is decreased to a monolayer that consists of one atomic layer of the metal Mo, sandwiched between two layers of the chalcogen atom, S [13]. The fact that these layered materials have strong internal covalent bonds but weak interlayered van der Waals bonds enables the process of cleaving the layers, also known as exfoliation. An example of such a method, which is also used in this study, is mechanical exfoliation using adhesive tape [4]. In thin films of TE materials, ZT can be enhanced due to the quantum confinement effect. This leads to enhanced density of states near the Fermi level [2,14,15]. The reported carrier concentration for undoped molybdenum disulfide (MoS_2_) is around $10^{13}\;{\mathrm{cm}}^{-3}$ [16]. The electrical conductivity of MoS_2_ can be increased through impurity doping, as described in the literature [17,18,19].

The Seebeck coefficient of MoS_2_ has been previously measured for 2D thin films of (Chemical vapour deposition) CVD deposited samples [20,21] to be ∼30 mV/K. For 2D materials, the quantum confinement of the electrons [13] make the materials to exhibit different electronics properties, as it has been reported for graphene [4] and other 2D materials [22]. The objective of this work was to present the results for micrometre thin films of MoS_2_, which do not exhibit the quantum confinement effect, but show relatively high Seebeck coefficient compared to the reported materials in nanometre scale. This work also presents the development of a measurement setup, which may be interesting and useful for studies of material properties of other potential thermoelectric materials and applications [6,7,8,9,10,11,12].

### 1.1. Seebeck Princple

A thermoelectric module is a device designed for the conversion of thermal energy into electrical energy. The conduction of heat through a material will generate a temperature gradient from the hot (which has higher kinetic energy due to higher thermal energy) to the cold side of the material. According to this principle, the cold side receives an excess flow of electrons from a higher-energy level (hot side) to the area of lower energy (cold side). Due to this electron diffusion, the hot side ends up having an excess of positive charges, or holes. This effect, known as the Seebeck effect, results in an electrical field proportional to the temperature gradient. The slope of the increase in the electrical field is called the Seebeck coefficient, as described by Equation (2).
(2)ΔS=ΔV/ΔT
where Δ*S* is the difference in the Seebeck coefficients of the materials (i.e., $\Delta S=\left|S_{A}-S_{B}\right|$, where *S_A_* and *S_B_* are Seebeck values of materials A and B). The reverse phenomenon, in which electrical energy is converted into thermal energy, is known as the Peltier effect [23]. 

### 1.2. Thermal Conductivity

There are a number of measurement techniques that can be used to estimate the thermal conductivity of different materials. The selection of the technique to be used depends on the shape and the size of the sample material [24,25,26]. 

The absolute technique, based on Fourier’s law of conduction, is a fundamental technique for the measurement of thermal conductivity. This technique can be employed to measure the thermal conductivity of thin samples by sandwiching the sample between the heat source and heat sink. When a steady-state power (*P*) is provided to the sample, the heat will flow through the sample and result in a temperature difference (Δ*T*) across the sample that can be measured by a PT-10K thermocouple. A steady-state temperature distribution over the sample is important for the validation of the measurement. The mathematical expression for Fourier’s law of conduction, in terms of thermal conductivity (*k*), is given in Equation (3) [27].
(3)k=(P×t)/(ΔT×A)
where *P* is the steady-state power, *t* is the thickness of the sample, *A* is the area of the sample and Δ*T* (Δ*T* = T2 − T1) is the temperature gradient across the sample. The steady-state power (*P*) can be calculated by multiplying the measured electrical voltage (*Vm*) and current (*Im*) given in Equation (4).
(4)$$P=V_{m}\times I_{m}$$

In Figure 1, the ZT of state-of-the-art TE materials versus temperature is illustrated [28]. It is possible to see that in an attempt to increase ZT, there is a trend of utilizing alloys rather than just the binary compounds. To be useful in TE applications, the material should have a ZT higher than 1. In this work, we present the ZT of n-type naturally doped MoS_2_ thin film. For this calculation, the Seebeck coefficient, the thermal conductivity and the electrical resistivity were measured. In the present study, the thermal conductivity measurement was done on the cross-plane of the sample, and the measurements of the temperature and electrical voltage were done in the plane of the sample. The details and the description of the measurement results and measurement setups will be presented in the following sections.

## 2. Methods

### 2.1. Seebeck Setup Design and Construction

A measurement setup was constructed for the estimation of the Seebeck coefficient on thin films in the plane direction. The setup is able to handle varying sample dimensions. The sample thickness could vary from a few micrometres up to a few millimetres, and the sample length could vary from 6 mm to 30 mm. The sample width could vary from 2 mm to 10 mm. A complete assembly of the setup is shown in Figure 2a. This measurement setup can handle a variety of samples of different dimensions compared to the presented measurement setup in [23]. This setup could be used to study the Seebeck coefficients of various materials of interest. The setup consists of hot and cold sides to generate a temperature gradient. A ceramic heater was used for heating the hot side. A Peltier element (PTE) was used to cool down one side of the sample. The PTE was installed between two aluminium plates, sample-holding and water-cooling sides, as shown in Figure 2a. When the PTE was connected to a power source, one side started cooling down and the other side started heating up. A water-based cooling system was constructed to cool down the hot side of the PTE, which allowed the cold side to decrease below 0 °C. In this way, a maximum 200 °C temperature difference (ΔT) was achieved for the measurement setup.

All mechanical parts of the setup were milled from a solid aluminium piece. The hot side was electrically insulated from the ceramic heater through a 0.5 mm thick wafer of aluminium oxide. In addition, we electrically insulated the whole setup using polytetrafluoroethylene (PTFE). Thermal conduction losses from the heater/cooler to the sample were minimized using thermal paste. The sample holders (hot and cold sides) were electrically insulated using Kapton tape (Eurostat, Brussels, Belgium) on all sides (the base, heater, and cooler sides). Additionally, the sample-holding clamps were covered with a thin film of Kapton tape to avoid any electrical connection between the sample and the measurement setup. The Seebeck coefficients for most of the materials are given with the reference of the platinum. Therefore, the contact pads were made of 0.4 mm thin platinum film and were extended with platinum wire to perform the electrical measurements (illustrated in green in Figure 2a).

### 2.2. Thermal Conductivity Measurement Setup

The setup used in the measurement of thermal conductivity is shown in the Figure 2b. It consisted of two rectangular aluminium blocks; one was connected to a heat source, and the other was connect to a heat sink. Resistive temperature sensors (PT-10K) (Sensor Technology IST AG, Wattwil, Switzerland) were installed inside the blocks with dimensions of 4 × 15 × 15 mm. To avoid heat losses, the heated block and the source were enclosed in multi-layered insulation material. Furthermore, the setup was placed into the vacuum chamber, and the pressure was reduced to 1.3 µ bar to minimize the convection losses. Steady-state power (P) was provided to the measurement setup, and the corresponding temperature was measured and logged.

### 2.3. Thickness Measurement

The thickness of the MoS_2_ sample was measured using a Mahr millitast 1083 instrument ( Mahr U.K. Plc, Milton Keynes, United Kingdom) with a resolution of 1µm. The instrument was installed on a vertical stand. The measurement was done with a flat tip of 3 mm width in different positions on the sample, and the average thickness (t) of the sample was estimated to be 170 µm.

### 2.4. XRD Measurement

The material composition of the MoS_2_ sample was characterized through a D2 Phaser XRD (manufacturer, city, country) from Bruker instruments USA. The thin film sample was placed on the sample holder, and the scan was performed from 2 theta angle of 10 to 50 degrees. The analysis of the thin film MoS_2_ through XRD is shown in Figure 3a.

## 3. Results and Discussion

### 3.1. Sample Preparation and X-ray Diffraction (XRD) Characterization

The sample of MoS_2_ was prepared by mechanical exfoliation of a molybdenite crystal supplied by the mining company Knaben in Norway. The exfoliation was conducted using the adhesive tape method described in [4]. The material composition of the sample was characterized using XRD, and the analysis of the thin film MoS_2_ is shown in Figure 3a. The characteristic 002 peak of MoS_2_, described in the literature card 1010993 [29], is visible at 14.6 degrees. The investigated sample was a single-crystal MoS_2_, and therefore, missing peaks were expected unless the crystal was oriented properly using a different instrument, such as a four-circle diffractometer [30]. This result is different compared to that of analysis on bulk powder [5], in which the crystallites have all possible orientations. However, the presented results are sufficient to say that the prepared material that was investigated was MoS_2_.

### 3.2. Measurement of the Seebeck Coefficient for Thin Film MoS_2_

The MoS_2_ sample was mounted on the setup, and the heater was ramped up by 0.25 °C/s, while the cooled side was kept at room temperature. A maximum temperature difference (∆T) of 140 °C was achieved. The temperature difference (∆T) and the induced Seebeck voltage across the sample were continuously measured using a temperature controller. The measurement results are shown in Figure 3b. The measured Seebeck voltage (left y-axis) and the calculated Seebeck coefficient (right y-axis) are plotted as a function of the temperature difference. Here, the Seebeck coefficient varies between −500 µV/K and −600 µV/K. A maximum Seebeck coefficient value of approximately −600 µV/K for MoS_2_ was estimated at a temperature difference of 40 °C. The negative sign shows that the MoS_2_ thin film sample was an n-type naturally doped semiconductor. For the p-type MoS_2_, the Seebeck value reported in the literature is 600 µV/K [31]. This investigation showed that the magnitude of the Seebeck coefficient of n-type MoS_2_ is similar to the p-type material, as was expected.

### 3.3. Measurement of Thermal Conductivity

Initially, the measurement setup was run without any sample to determine the heat loss through the setup (i.e., temperature difference between heat source and heat sink without any sample). For this purpose, steady-state power (P) was provided to the measurement setup, and the corresponding temperatures of both sides were measured and logged. This measured temperature difference gives the temperature loss (ΔT_losses) of the setup. The measured temperature difference is plotted in Figure 4 and was estimated to be about 2.53 ± 0.003 °C.

The MoS_2_ sample was cut into a square-shaped sample with an area (A) of approximately 123 mm^2^ and measured thickness of 170 µm. The sample was sandwiched between the heat source and the heat sink, as shown in Figure 2b, and placed into a vacuum chamber with a pressure of 1.3 µ bar. An electric ceramic-based heater connected with a power source provided the steady-state power to the heating side. The provided power was estimated to be about 1.42 W using Equation (4). The steady-state equilibrium condition of the temperature across the setup was achieved by running the measurement for a period of several hours, in both cases, the temperature was almost constant for several hours, and a part from data are presented. With the MoS_2_ sample, the measured temperature (ΔT_meas) was estimated to be 9.99 ± 0.003 °C, as presented in Figure 4. This corresponds to an actual temperature difference (ΔT_actual) of 7.45 °C across the sample, determined by subtracting the ΔT_losses from ΔT_meas. Using Equation (3), the thermal conductivity of the MoS_2_ was calculated to be 0.26 ${\mathrm{Wm}}^{-1}K^{-1}$. In comparison, the calculated thermal conductivity of the exfoliated MoS_2_ sample was lower than for bulk material. As reported in [32], bulk mono-crystals (001) of MoS_2_ have a thermal conductivity in cross-plane directions equal to 2 Wm^−1^ K^−1^. 

### 3.4. Calculation of ZT for MoS_2_

The sheet resistance of the MoS_2_ sample was estimated to be 1620 Ω/⬜ (ohm/square)using a four-point probe, which enabled minimizing the contact resistance of probes and cables. By multiplying the sheet resistance with the thickness of the sample, the electrical resistivity of the sample was estimated to be 29 Ω cm. The figure of merit (ZT) was calculated to be 1.99 × 10^−4^ by substituting the values of electrical resistivity, thermal conductivity and the Seebeck coefficient in Equation (1). According to Equation (1), to achieve a high ZT value, a material should have a high Seebeck coefficient and low thermal and electrical resistivity. Even with a very high Seebeck coefficient, high electrical resistivity of a naturally doped n-type MoS_2_ sample affects the ZT value. However, through impurity doping [17,18,19], the electrical conductivity of the MoS_2_ can be increased, which affects the ZT of MoS_2_. For instance, to obtain a ZT of MoS_2_ to the magnitude of 1, the electrical resistivity value should be equal to 5.48 × 10^−3^ Ω cm, that is, 5000 times lower compared to the current value. This material could have wide possible applications in TE generators in the future.

## 4. Summary and Conclusions

The thermal conductivity and Seebeck coefficient measurements of a naturally doped micrometre-thick MoS_2_ sample are presented in this work. The MoS_2_ sample was obtained from a crystal of molybdenite through an exfoliation method. Initially, the film of MoS_2_ was analysed through XRD, and the results showed that the investigated sample was the mineral MoS_2_. Further characterization of the MoS_2_ sample was performed using two different measurement setups that were designed and built to measure both the Seebeck coefficient and the thermal conductivity. These measurement setups are easy to build and can be used to characterize other thermoelectric materials as well. The Seebeck coefficient was estimated to be −600 µV/K at a temperature difference of 40 °C. The negative sign shows that the MoS_2_ film sample was an n-type naturally doped semiconductor. This investigation showed that the magnitude of the Seebeck coefficient of n-type MoS_2_ is similar to the p-type material, as was expected. The thermal conductivity of the sample was estimated to be 0.26 ${\mathrm{Wm}}^{-1}K^{-1}$ using the Fourier’s law of conduction. The electrical sheet resistance of the MoS_2_ sample was estimated to be 1620 Ω/□ (ohm/square), which results in an electrical resistivity of 29 Ω cm. The figure of merit (ZT) of MoS_2_ was estimated to be approximately 1.99 × 10^−4^. In order to achieve a high ZT value for MoS_2_, the electrical resistivity should be very low compared to the current value.

## Figures and Tables

**Figure 1 materials-12-03521-f001:**
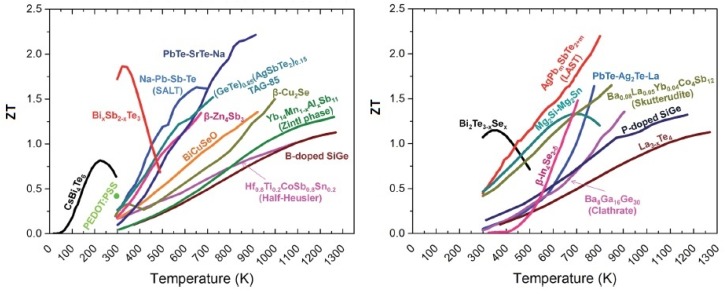
ZT values for some bulk thermoelectric materials as a function of temperature: (left) p-type TE materials and (right) n-type TE materials. Reproduced from [28] with permission from The Royal Society of Chemistry.

**Figure 2 materials-12-03521-f002:**
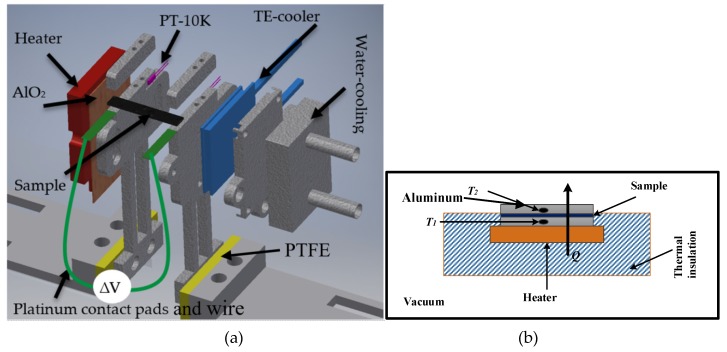
(**a**) A schematic view of the Seebeck measurement setup. The setup consists of two sides, heating side along with heater attached (red coloured), and cold side along with TE-cooler (blue coloured). All grey coloured parts are aluminium, the yellow coloured parts are PTFE, and orange coloured part is aluminium oxide. In (**b**), a schematic view of the thermal conductivity measurement setup, where heater and heating side is enclosed in multi-layered insulation, sample is sandwiched between the two aluminium pieces, and temperature sensors are installed inside the aluminium, and the whole setup was placed in the vacuum.

**Figure 3 materials-12-03521-f003:**
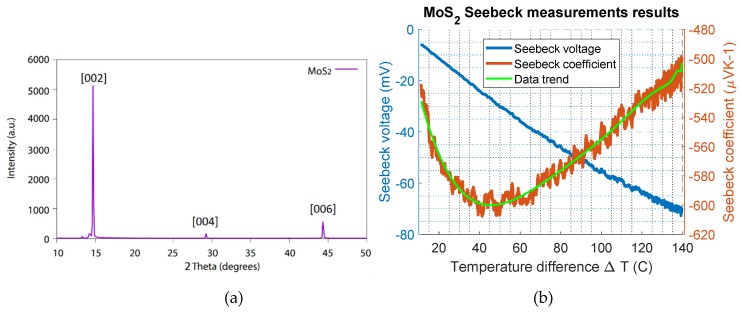
(**a**) XRD measurement data of MoS_2_, the figure showing peaks at 14.7, 29.2 and 44.3 degrees and in (**b**) the Seebeck measurement results are given, the Seebeck coefficient of MoS_2_ plotted as a function of the temperature difference on the right y-axis; the Seebeck voltage is given on the left y-axis.

**Figure 4 materials-12-03521-f004:**
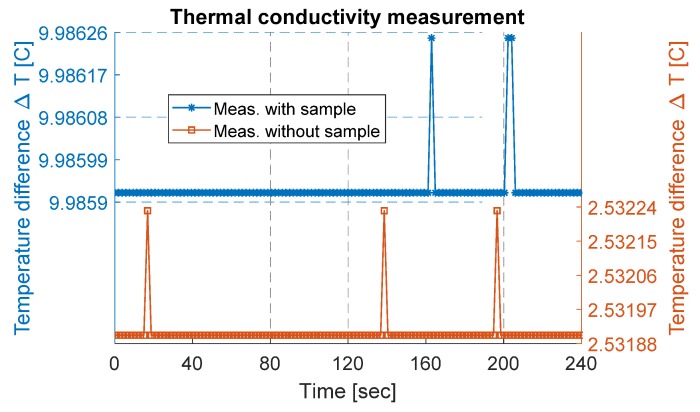
Thermal conductivity measurement data plotted for a few minutes. On the right y-axis, measurement data without a sample are plotted, while on the left y-axis, measurement data with a sample are plotted.

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
