# Peer review of "Thermoelectric Properties of n-Type Molybdenum Disulfide (MoS2) Thin Film by Using a Simple Measurement Method"

_materials, 2019, doi:10.3390/ma12213521_

Round 1

Reviewer 1 Report

Manuscript ID: materials-606991

Title: Thermoelectric properties of n-type molybdenum disulfide (MoS2) thin film

Comments:

The authors report on thermoelectric properties of ~170um thick MoS2 flake achieved by mechanical exfoliation method. Even though the manuscript is well written, it doesn’t include scientific impact for materials journal. The experimental data are expected as well. The manuscript needs more a group of data and scientific impact. The reviewer rejects this manuscript.

Author Response

In this paper, we have presented the thermoelectric properties for n-type MoS2 by using a developed measurement setup. To characterize a material for its thermoelectric properties, the figure of merit (ZT) was calculated. We have measured the electric conductivity, Seebeck coefficient, and measured its thermal conductivity. For this purpose, two measurements setup were established—which can be used to measure thermoelectric properties of other TE material of interest. To our knowledge, the thermoelectric properties of n-type MoS2 have not been published yet. By presenting the thermoelectric properties of n-type MoS2 and proposing a measurement method is sounds scientific enough as there is a need to find materials with good thermoelectric properties and although MoS2 did not show so spectacular figure of merits the goal was to show the construction of the equipment as well.

So, we think that his request is to general and not possible to address with this short deadline. Thus, we request to the editor and review that please re-evaluate the comments before rejecting this paper.

Reviewer 2 Report

It is a great article dealing with Thermoelectric properties of n-type molybdenum disulfide (MoS2) thin film.

I have several recommendations for gaining more interest to the paper by a broad range of researchers, dealing with the development of various classes of thermoelectric materials other than MoS2.

For those, I would recommend to include in the introduction, a sentence in the form of, many thermoelectric materials are being explored for power generation applications, such as germanium telluride with the following reference:

Hazan, N. Madar, M. Parag, V. Casian, O. Ben-Yehuda and Y. Gelbstein, Effective electronic mechanisms for optimizing the thermoelectric properties of GeTe-rich alloys, Advanced Electronic Materials 1(11) 1500228 (2015).

Silicides, with the following reference:

Nieroda, J. Leszczynski and A. Koleznyski, Bismuth doped Mg2Si with improved homogeneity: synthesis, characterization and optimization of thermoelectric properties, Journal of Physics and Chemistry of Solids 103 147-159 (2017)

and half-Heuslers with the following reference:

Tanja Graf, Peter Klaer, Joachim Barth, Benjamin Balke, Hans-Joachim Elmers and Claudia Felser, Phase separation in the quaternary Heusler compound CoTi(1-x)MnxSb- a reduction in the thermal conductivity for thermoelectric applications, Scripta Materialia 63 1216-1219 (2010).

Following taking into accounts the minor revisions specified above I will be glad to recommend on acceptance of the manuscript.

Author Response

Response: We have revised the manuscript according to the reviewer comments and changes are highlighted in the manuscript attached below. 

Round 2

Reviewer 1 Report

Manuscript ID: materials-606991R

Title: Thermoelectric properties of n-type molybdenum disulfide (MoS2) thin film by using a simple measurement method

Comments:

The authors report on thermoelectric properties of ~170um thick MoS2 flake achieved by mechanical exfoliation method. It is true that the manuscript is well written. According to the authors’ respond report, this manuscript is to propose a (new) measurement method for thermoelectric characteristics. However, there is no specific mention on the abstract and conclusion (pls, add it). And, pls include specific information on the proposed method compared to the conventional one (differences, advantages etc.) in the right part. Fig. 3(a) shows 3 peaks and the authors add plane orientations on the Fig. What are the others except (002) at 14.66 degrees.

Author Response

Dear Reviewer,

Thank you very much for being a reviewer of our manuscript and giving us time.

Response: The authors have updated the manuscript according to the reviewer's comments and the changes are highlighted in the manuscript. We have written about the measurement setup in the abstract and in the conclusion. Our measurement setup was also compared with another one, which was built up for the same purpose. The authors have also updated the figure 3 with the Miller indexes for the XRD measurement on the thin MoS2 changing the legend slightly (page 5). Other peaks common to MoS2 are not showing in this measurement because of the orientation of the atoms in the crystal of the sample we tested. Since it is just one flake and we measured in only one direction, no extra peaks were observed. “About the other expectation of peaks at 14.66 degrees” comment, the author has no knowledge of other materials. To our knowledge and experience, these results are common for Molybdenite, the ore of MoS2.

We would like to thank the reviewers and author services from the MDPI journal and hopefully this time our answers satisfy the reviewers and we can proceed with the publication of our work.

Best regards,
Mittuniversitetet

Dr. Shakeel Ashraf